# Characterization of Response Frequency for the Pressure-Sensitive Paint Based on Fluorine-Containing Polymer Matrixes Using Oscillating Sound Wave Technique

**DOI:** 10.3390/s20216310

**Published:** 2020-11-05

**Authors:** Yun Shao, Liusheng Chen, Qi Liao, Heng Jiang, Zhitian Liu, Xigao Jin, Limin Gao

**Affiliations:** 1Institute of Materials for Optoelectronics and New Energy, School of Material Science and Engineering, Wuhan Institude of Technology, Wuhan 430205, China; shaoyun_wit@163.com; 2State Key Laboratory of Polymer Physics and Chemistry, Institute of Chemistry, Chinese Academy of Sciences, Beijing 100190, China; chenls@iccas.ac.cn (L.C.); qiliao@iccas.ac.cn (Q.L.); 3School of Power and Energy, Northwestern Polytechnical University, Xi’an 710072, China; hengjiang@mail.nwpu.edu.cn

**Keywords:** pressure-sensitive paint, fluorine-containing polymer, response frequency, oscillating sound wave

## Abstract

Five kinds of new homo-polymer and copolymers of methacrylate containing a fluorine ester group were synthesized and used for the binder of pressure-sensitive paint (PSP)to ensure the good compatibility between luminophore (Pt(II) meso-tetra (pentafluorophenyl) porphine (PtTFPP)) and polymer binder. In the work, we were concerned with how the structure of thesepolymers containing fluorine, especially the various ester group structure, affects the response frequency of PSP using oscillating sound wave technique. The results showed that the pressure sensitivities (Sp) of these PSP samples containing different polymers, exhibit some difference. The length of ester chain on the methacrylatepolymer affects the response frequency of PSP sensor layer composed of the polymer. The longer the chain length of the ester group, the higher the response frequency of the PSP sensor layer quenching by oxygen.

## 1. Introduction

The luminescence imaging combined with pressure-sensitive paint (PSP) is a novel method for measuring surface air pressure distribution on aerodynamic models in wind tunnel and other environments [1,2,3,4,5,6,7,8,9,10]. PSP is advantageous because it can be applied in the situations where conventional instrumentation over traditional electrochemical sensors is difficult, or even impossible to apply. The PSP is comprised of the luminescent dye molecules dissolved or dispersed in polymer matrix. The working principle of this technique is when the PSP sensor film is illuminated at a wavelength which could be absorbed by the dye, it becomes electronically excited and then fluorescence or phosphorescence emits at a longer wavelength. Oxygen is a powerful quencher for this emission, as oxygen diffuses into or out of the sensor layer, the extent of quenching changes and then could be detected by the change in luminescence intensity or luminescence decay rate of the dye in the matrix. The oxygen quenching process depends directly on the concentration of oxygen presented in the test gas. The higher the concentration of oxygen in the test gas, the more energy will be dissipated through the oxygen quenching process.

In recent years, it has been a growing interest in measuring these unsteady pressure fluctuations [11,12,13,14]. In particular, unsteady aerodynamic issues in the compressors such as rotating stall, surge, rotor/stator interactions and flutter need to be measured and studied. Up to recent times, however, PSP has been limited to be used in steady flow fields. Typically, conventional PSP has response times on the order of a few hundred milliseconds, making it impractical for use in unsteady turbomachinery flow fields.

The relation of frequency (f, Hz) and time (t, sec) may be expressed in f = 1/t, therefore the response time of PSP can be calculated from the response frequency of these PSP. The response frequency or response time of oxygen quenching of PSP is dependent on two properties of the paint: the lifetime of the luminophores, and the oxygen diffusivity in the binding polymer. Since the lifetime for many luminophores is on the order of a microsecond, the oxygen diffusivity of the paint is the limiting factor in the response frequency of the PSP. There sponse models of PSP demonstrate that the time scale of the oxygen diffusion process within the paint binder is dependent upon the thickness of the binder and the oxygen diffusivity of the binder material [15]. This concept is expressed as
(1)$$\tau_{\mathrm{diff}}\propto \frac{h^{2}}{D_{m}}$$
where h is the binder thickness, D_m_ is the oxygen diffusivity of the binder layer and τ_diff_ is the diffusion timescale. From this expression, it is clear that the response frequency of the paint can be improved by decreasing the paint thickness, or by increasing its oxygen diffusivity. Therefore, research efforts should be focused on improving the oxygen diffusion characteristics of paint binders.

Many of the problems and limitations of the PSP technique are associated with the choice of dye and polymer matrix. At present, many works focused in the probe choice [3,4,16,17,18,19], but only a few literatures reported about how the detailed polymer structure within PSP affects the oxygen permeability [17]. Gas diffusion and permeation in polymers is normally measured by a transport experiment involving a thin polymer membrane as first described by Barrer [20]. Oxygen permeability for PSP is related to the response frequency or response time, the higher the permeability of oxygen in polymer, the higher the response frequency of PSP. There is a need for a new calibration technique that demonstrates the full extent of the capabilities of PSP. At the present paper, the response frequency of PSP is characterized with a new dynamic calibration method using anoscillating sound wave technique [21,22,23]. The loudspeaker calibration setup can be driven to large frequencies, on the order of tens kilohertz. The ability to determine the input waveform allows the researcher to obtain phase delay and amplitude attenuation data in the frequency domain.

Here, we used platinum (II) meso-tetrakis (pentafluorophenyl) porphyrin (PtTFPP) as the luminophore due to its high photostability and high emission quantum efficiency [24]. In this work we synthesized a new homopolymer and four kind of copolymers of methacrylate containing fluorine ester group used for the binder of paint to ensure the compatibility between luminophore and polymer binder. We are concerned with how the structure of a series of polymers containing fluorine, especially the ester length, affects the response frequency of PSP.

## 2. Materials and Methods

### 2.1. Materials

#### 2.1.1. Chemical Reagents

Hexafluoroisopropyl methacrylate(HFIPMA) was obtained from Apollo Scientific Ltd. 2,2,3,3-Tetrafluoropropyl methacrylate(TFPMA), 2,2,2-trifluoroethyl methacrylate(TFEMA), 1*H*,1*H*,2*H*,2*H*-*Nonafluorohexyl* methacrylate(NFHMA) and 2,2,3,3,3-pentafluoropropyl methacrylate(PFPMA) were gotfrom TCI and these five kinds of monomers were used without any further purification. Lauroyl peroxide (LRPO, 97%) was recrystallized in methanol before using. Benzotrifluoride (99%) was purchased from J&K Scientific (Beijing, China). PtTFPP was commercially available from Frontier Scientific Inc. Methanol (99.8%) (Logan, UT, USA) and petroleum ether were obtained from Beijing Chemical Works (Beijing, China). Scheme 1 presents their chemical structures.

#### 2.1.2. Experimental Instruments

All steady-state excitation and emission measurements were carried out on F-4500 (Hitachi, Japan) fluorescence spectrophotometer. The excitation and emission spectra were measured by the front-face technique with an excitation wavelength at 400 nm and emission wavelength at 650 nm, respectively. The glass transition temperatures (T_g_) of the polymers were determined using a differential scanning calorimeter (TA-2000, Japan) with a scan rate of 10 °C/min. The infrared spectra were measured on a Tensor 27 (Bruker, Yokohama, Japan) spectrometer. Size exclusion chromatography (SEC) characterization was performed on the Breeze 2 HPLC (Water Ltd., Hongkong, China) instrument with tetrahydrofuran (THF) as the mobile phase at the flow rate of 1 mL/min. The SEC data were calibrated with polystyrene standards ranging in the molecular mass from 28.7 to 1240 kg mol^−1^. 

The static properties of the PSP samples were measured in a temperature-controlled pressure chamber (Scheme 2a). Excitation was provided by a 300 W Xe lamp (C 4338, Hamamatsu Photonics, Hamamatsu, Japan) equipped with a 400–550 bandpass filter. The emission from the sample was detected by a cooled CCD camera (C 4880-50-26W, Hamamatsu Photonics; 1000 3 1018 pixels, 16-bit intensity resolution) with a 590–710 nm bandpass filter. The luminescence intensities I(T,P) were collected at test pressures (P) ranging from 5 to 120 kPa and at temperatures (T) ranging from 0 to 60 °C at 10 degree intervals. The dynamic characteristics of the PSP samples were measured on the domestic dynamic pressure optical calibration system (DPOCS, Scheme 2b), developed by Northwestern Polytechnical University (NPU). It is comprised of a dynamic pressure chamber, an ultraviolet (UV) light with 405 ± 20 nm, a photo multiplier tube (PMT) with a red filter and other auxiliary sub-sets. A series of sine pressure wave from 0.4~80 kHz can be generated by the pressure chamber with a 1.93 kPa maximum pressure [25,26].

### 2.2. Synthesis of Methacrylate Polymerswith Fluorine-ContainingEster Group

Polymers were prepared following the method of Puklin et al. [27]. In the typical copolymerization procedure, HFIPMA and PFPMA in a 1:1 molar rate, were added into the three-neck flask equipped with a mechanical stirrer, a reflux condenser and a temperature controller, meanwhile LRPO with 0.15% of monomers mass, benzotrifluoride in twice the volume of the monomers were added into the system. Nitrogen gas was introduced into the solution for 1 h. The solution was heated at 78 °C for 48 h and then at 100 °C for 1 h, and finally cooled to ambient temperature. The polymer was reprecipitated in methanol, filtered and dried in vacuum oven at 45 °C for 24 h to give poly(HFIPMA-co-PFPMA) as a white solid.

Similarly, the other fluorine-containing polymers were synthesized according to the procedure as mentioned above. Other monomers e.g., TFEMA, TFPMA and NFHMA were copolymerized with HFIPMA individually to obtain three kind of copolymers referred as P(HFIPMA-co-TFEMA), P(HFIPMA-co-TFPMA) and P(HEIPMA-co-NFHMA), respectively. P(HFIPMA) is a homopolymer of HFIPMA. In these five kinds of polymers, P(HFIPMA-co-TFEMA) and P(HFIPMA-co-TFPMA) were reprecipitated in petroleum ether, and other polymers were reprecipitated in methanol.

### 2.3. Preparation of PSP Samples

The five polymers prepared were dissolved individually in benzotrifluoride with the concentration of 120.0 g/L, and then PtTFPP was added in the polymer solution in the concentration of 4 × 10^−5^ mol/L. The mixed solution was sprayed onto a clean aluminum slides (25 × 25 mm^2^) using a conventional air brush, and then the solvent was evaporated slowly in the dark at room temperature, after which the films were placed in a vacuum for 5 h at 50 °C. The PSP sample containing PtTFPP in various polymer matrices were represented as Co(PFPMA), Co(TFEMA), Co(TFPMA), Co(NFHMA) and P(HFIPMA), respectively.

## 3. Results and Discussion

### 3.1. Characterization and Basic Structure and Properties of the Polymers

As an illustration, Figure 1 shows the IR spectra of P(HFIPMA-co-PFPMA) and P(HFIPMA). The absorption bands of IR presented at 2976, 1485, 1386(C-H_3_), 2923, 1461(C-H_2_), 3001, 1357(C-H influenced by F), 1776(C=O influenced by F), 1292, 1236(C-F), 1200(O-C=O), 1110(C-O-C influenced by F) and 1020 cm^−1^(C-O-C), respectively. It can be observed that there are two kinds of characteristic peaks of C-O-C in the infrared spectrum for P(HFIPMA-co-PFPMA) and only one kind of characteristic peak of C-O-C for P(HFIPMA), which proven that there are two kinds of C-O-C in P(HFIPMA-co-PFPMA), that means the successful synthesis of P(HFIPMA-co-PFPMA). The spectra of other copolymers are similar. Based on the single glass transition behavior in the DSC curves, it is also referred to the copolymer structure.

Table 1 shows the essential physical parameters of these polymers. The T_g_ of the copolymer containing-NFHMA is the lowest (61 °C) in all of these polymers, the T_g_ of P(HFIPMA-co-TFEMA) is the highest (81 °C), it means that as the alkyl substituent is increased in length, the glass transition temperature of the polymer decreased. The tendency is similar to that reported from Winnik et al. [28]. They studied the relation between T_g_ and the substituent on poly(n-alkyaminothionylphosphazenes)(CnPTP), and the result indicated that as the alkyl substituent was increased in length, the glass transition temperature of the polymer decreased.

### 3.2. Measurement of Fluorescence Spectra

Figure 2 is the emission spectra of PtTFPP in N_2_ and air environment, respectively. It shows that the emission intensity of PtTFPP was significantly quenched by oxygen in air environment and then that is very low. The excitation and emission spectra (Figure 3a,b) of individual polymer/PtTFPP films were obtained in N_2_ environment, respectively. Based on Figure 2 and Figure 3, it could also be found that the excitation and emission spectral sharps and peak positions of PtTFPP in different polymers are the identical. The results indicated that the interactions between luminescent probe PtTFPP and different polymers incorporated probe are the same [29].

### 3.3. Effect of Polymer Composition on the Pressure Sensitivity of PSP

Based on the photophysical principle of the PSP, the relationship between the release of luminescent energy from the excited luminophore and the concentration of quencher (oxygen or air pressure) can be characterized by the Stern-Volmer (S-V) Equation (3):(2)$$\frac{I_{\mathrm{ref}}}{I}=A+B\;\frac{P}{P_{\mathrm{ref}}}$$
where P_ref_ and I_ref_ are the reference pressure and emission intensity at the reference pressure, respectively. P and I refer to the test pressure and luminescent intensity at the test pressure, respectively. The pressure sensitivity of the PSP can be obtained from S-V plotted (Figure 4) from I_ref_/I against P/P_ref_ measured in a temperature-controlled pressure chamber (Scheme 2a). Table 2 also presents the relation between the polymer composition and pressure sensitivity of the PSP. The results show that the pressure sensitivities (Sp) of these PSP samples consisted of different polymers exhibit some difference.

### 3.4. Effect of Polymer Composition on Response Frequency

The working principle of PSP is based on the oxygen diffuses into or out of the PSP sensor layer and then the luminescence intensity emitted from sensor layer changes. The diffusion motion of small molecules such as the permanent gases (oxygen, nitrogen, carbon dioxide, etc.) can occur in solid polymer at the temperatures well below its glass transition temperature. It is believed that the motion of these gas molecules through the polymer matrix is related with the formation of free volume which is assisted by small-scale motion of polymer segments or the free movement of the groups attached to the side chains.

Figure 5 shows the raw data of the response frequency (F) of PSP based on the five kinds of polymer matrixes. Table 3 lists the response frequency, and response time (T) calculated from the response frequency results. It can be found that response frequency of Co(NFHMA) is the highest in 20 kHz, and that of Co(TFEMA) is the lowest in 9.6 kHz. For other samples, e.g., Co(TFPMA), Co(PFPMA) and P(HFIPMA), the structures of their side-ester chains are -O-CH_2_-CF_2_-CHF_2_, -O-CH_2_-CF_2_-CF_3_ and -O-CH-(CF_3_)_2_, respectively, compared these chemical structures with that of -O-(CH_2_)_2_-(CF_2_)_3_-CF_3_ on Co(NFHMA) and -O-CH_2_-CF_3_ on P(TFEMA), their response frequency are 19 kHz, 18 kHz and 16 kHz, respectively. The results can be explained from the relation between the linkage structure and free volume in the polymer matrixes. The important concept, common to all free volume theories, is that the diffusion occurs in polymers through free volume obtained by minor displacements of side groups or segments of the chain but without net translational movements of the center of mass of the polymer. There is a longer and flexible ester chain containing -O-(CH_2_)_2_-(CF_2_)_3_-CF_3_ on the ester group in Co(NFHMA), and its free volume may be created by the chain segment movement on the ester group. Otherwise, there is a shorter side-chain containing -O-CH_2_-CF_3_ on the ester group in Co(TFEMA), and its free volume may be created by the rotation of the ester group. Guillet [30] studied the “transition temperatures” of molecular motion within polymers using luminescence measurements and reported that the transition temperatures of –C-C- side-chain segment movement and ester group rotation for polymer chains were around 163 K and 245 K, respectively. They found that the activation energy associated with diffusion above particular transitions increases with the transition temperature. This is in good agreement with the free-volume theory developed early for diffusive motion in solid polymers. Therefore, the activation energy of the side-chain segment movement for Co(NFHMA) is lower than that for Co(TFEMA), and the response frequency of Co(NFHMA) is higher than that of Co(TFEMA). It can be understood that the longer the ester group chain length, the response frequency of PSP is higher. Therefore, it indicates that the chain length of ester group on the methacrylate polymers affects the response frequency of PSP sensor layer composed of the methacrylate polymers.

## 4. Conclusions

The response behaviors of the PSP based on fluorine-containing polymer matrixes were investigated in this article. Especially, we focused the effect of polymer binder structure on the response frequency of PSP. For this consideration, five kinds of methacrylate polymers with different fluorine-containing ester groups were synthesized, and then the polymers were composed with the luminescent dye of PtTFPP individually. The response frequency of the five PSPs were measured by using oscillating sound wave technique, and the results show that the polymer structure of various fluorine-containing ester group affects significantly on the response frequency. That could be explained based on free volume theory, i.e., for the polymer with a longer fluorine-containing ester chain on methacrylate, its free volume is created by the chain segment movement, then the activation energy of the chain segment movement of the ester group decreased, which shows the increasing in the response frequency of PSP.

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
