# Peer review of "Characterization of Response Frequency for the Pressure-Sensitive Paint Based on Fluorine-Containing Polymer Matrixes Using Oscillating Sound Wave Technique"

_sensors, 2020, doi:10.3390/s20216310_

Round 1

Reviewer 1 Report

As follows are some comments/suggestions to the authors for improving/revising the manuscript.

(1) Line 87-94: “Hexafluoroisopropyl methacrylate(HFIPMA) was obtained from Apollo Scientific Ltd.  2,2,3,3-Tetrafluoropropyl methacrylate(TFPMA), 2,2,2-trifluoroethyl methacrylate(TFEMA),  1h,1h,2h,2h-nonafluorohexyl methacrylate(NFHMA) and 2,2,3,3,3-pentafluoropropyl methacrylate(PFPMA), and Pt(II) meso-tetra(pentafluorophenyl)porphine(PtTFPP)”.  

For better readability, please provide a figure that presents the structure of the above chemicals.

(2) line 89 and 134: “1h,1h,2h,2h-nonafluorohexyl” should be corrected as “1H,1H,2H,2H-Nonafluorohexyl”.  There’re quite a few of other typo errors in this manuscript, please carefully double-check your manuscript.

(3) Lines  92 and 124: for consistency, please use the same name for benzotrifluoride and trifluorotoluene.

(4) lines 134-135: “HFIPMA was self-polymerized to get a homopolymer of P(HFIPMA).”  The term “self-polymerize” is confusing.  Do you mean that no initiator is used in this polymerization? Please provide more details of the polymerization conditions for clarity.

(5) Lines 54-61: “film thickness is one of the key factors that influence the PSP properties”.  For a target thickness, the preparation method should be consistent and reproducible.  However, in the experimental section, it seems that the film is prepared manually spray-coated with a conventional air brush.  No detailed description is provided to get the target dry film thickness, 30-40 micrometers.

(6) Lines 150-157:  for the conclusion of “successful synthesis of P(HFIPMA-co-PFPMA”, the FTIR analysis is not sufficient as evidence to confirm that it is a copolymer, because a mixture of the homopolymers of PHFIPMA and PPFPMA also has the peak at 1020 cm-1.  NMR would provide more scientific information on the sequence of the monomeric structure in the polymers than FTIR. 

(7) Line 109-120:  For better readability, it is suggested you provide diagrams/figures or video clips (as a supplementary information) for the setup and procedure for the measurement of the PSPs.

Reviewer 2 Report

The manuscript investigates how the polymers used as pressure-sensitive paints affect the luminophore response using an original wave sound technique.

My recommendation is for publishing after minor revision, including some improvements of English language.

I list below some observations for the authors:

The polymers were synthesized using radical polymerization or copolymerization. By this technique it’s difficult to obtain strictly linear polymers and copolymers (Abstract, raw 13, Introduction, page 2, raw 80). In my opinion the term “linear” must be deleted. Also in Abstract, raw 18: the phrase “The results showed…” must be reformulated because is ambiguous. I would suggest: “The results show that the pressure sensitivities (Sp) of these PSP samples, containing different polymers, exhibit some differences.” Page 1, r 27: I would say “Using pressure-sensitive paints (PSP) is a well-developed optical technique…” as PSP can’t be a technique, but their use is. Page 1, r 30: “PSP is advantageous because…” instead of PSP is advantages… Page 3, r 124: “…meanwhile LRPO with monomers mass of 0.15%...” what is the meaning here? Page 4, r 135: “From these five kinds…” Page 5, r 174: “…are identical.” Page 6, eq. 2: A and B are some fitting parameters or do they have a significance? Also, please explain how the pressure sensitivity is obtained from plotting the mentioned equation. I consider useful to show at least one plot Iref/I vs Pref/P to see how well the Stern-Volmer equation fit the experimental data. Page 6, r 188: “The results….” Please reformulate (see 1 in this list) Page 6, r 197: What is “liberational movement”? Figure 4: What is Kulite? Also the legend for fig. 4 contains: …”under the frequency of (a) 4K, (b) 16 K…” What is K as a unit measure for frequency? Table 3: Please offer more details about how the response frequencies were obtained from the raw data.
